# Decay and damage of therapeutic phage OMKO1 by environmental stressors

**Michael Blazanin**[1]*, **Wai Tin Lam**[1], **Eli Vasen**[1], **Benjamin K. Chan**[1], **Paul E. Turner**[1,2]

**1** Department of Ecology and Evolutionary Biology, Yale University, New Haven, CT, United States of America, **2** Program in Microbiology, Yale School of Medicine, New Haven, CT, United States of America

☉ These authors contributed equally to this work.
* mike.blazanin@yale.edu

## Abstract

Antibiotic resistant bacterial pathogens are increasingly prevalent, driving the need for alternative approaches to chemical antibiotics when treating infections. One such approach is bacteriophage therapy: the use of bacteria-specific viruses that lyse (kill) their host cells. Just as the effect of environmental conditions (e.g. elevated temperature) on antibiotic efficacy is well-studied, the effect of environmental stressors on the potency of phage therapy candidates demands examination. Therapeutic phage OMKO1 infects and kills the opportunistic human pathogen *Pseudomonas aeruginosa*. Here, we used phage OMKO1 as a model to test how environmental stressors can lead to damage and decay of virus particles. We assessed the effects of elevated temperatures, saline concentrations, and urea concentrations. We observed that OMKO1 particles were highly tolerant to different saline concentrations, but decayed more rapidly at elevated temperatures and under high concentrations of urea. Additionally, we found that exposure to elevated temperature reduced the ability of surviving phage particles to suppress the growth of *P. aeruginosa*, suggesting a temperature-induced damage. Our findings demonstrate that OMKO1 is highly tolerant to a range of conditions that could be experienced inside and outside the human body, while also showing the need for careful characterization of therapeutic phages to ensure that environmental exposure does not compromise their expected potency, dosing, and pharmacokinetics.

## Introduction

Widespread use of antibiotics–particularly in human therapy and animal agriculture–has selected for the evolution of multi-drug resistant bacterial pathogens, commonly associated with poorer prognosis and higher morbidity in human infections [1, 2]. One alternative or complementary approach to treating bacterial infections with chemical antibiotics is bacteriophage therapy [3], where bacteria-specific viruses with lytic replication cycles are used to kill (lyse) target bacterial cells. Phage virion particles, like antibiotic molecules, can be sensitive to environmental conditions. So, just as past work has assessed the sensitivity of chemical antibiotics to conditions like high temperature [4–7], it is important to observe how phage particle

**Data Availability Statement:** All data and analyses are available at https://github.com/mikeblazanin/tin-omko1. Cleaned data and analysis code is also archived at https://doi.org/10.5061/dryad.k3j9kd595.

**Funding:** We disclose financial support for Wai Tin Lam from the Chinese University of Hong Kong. The funders had no role in study design, data collection and analysis, decision to publish, or preparation of the manuscript.

**Competing interests:** Paul E. Turner discloses a financial interest in Felix Biotechnology, Inc., a phage therapeutic company with first rights to use patents resulting from this work. Paul Turner sits on the Board of Directors of Nextbiotics. This does not alter our adherence to PLOS ONE policies on sharing data and materials.

survival and growth is altered by environmental stressors. In particular, understanding the physiological limits of phage particles has implications for both the characterization and application of therapeutic phages. Here, we used a therapeutic phage's responses to salinity, urea concentration, and high temperatures as a model of virion decay and damage by exposure to environmental stressors.

The need for novel approaches to treat bacterial pathogens varies greatly between infections: some strains of bacterial pathogens remain treatable with conventional antibiotics, while others show resistance across multiple drug classes, sometimes to all currently-approved antibiotics [8]. One of these multi-drug resistant bacterial pathogens is *Pseudomonas aeruginosa*, which has been identified by the World Health Organization as a high-priority threat to human health [9]. *P. aeruginosa* is a widespread gram-negative opportunistic pathogen that is commonly found in both natural habitats (e.g., soil, fresh water) and artificial environments (e.g., sewage, households, hospitals, and contaminated medical equipment) [10–13]. For these reasons, *P. aeruginosa* is frequently encountered by humans and causes urinary-tract and respiratory infections in immunocompromised individuals, as well as fouling surgically-implanted materials and devices [14, 15]. Individuals with cystic fibrosis (CF), non-CF bronchiectasis, and chronic-obstructive pulmonary disease (COPD) are especially vulnerable to lung infections caused by *P. aeruginosa*. Treatment of *P. aeruginosa* with chemical antibiotics is often ineffective, both because the bacteria readily form biofilms that limit the penetrance of antibiotic molecules, and due to multi-drug efflux (Mex) systems: protein complexes that actively remove various types of antibiotics from the cell [16–18].

As an alternative or complementary approach to treating *P. aeruginosa*, phage therapy has many attractive advantages, such as the ability for the phage 'drug' to self-amplify within the infection site. However, phage therapy also has some disadvantages, most notably the rapid evolution of bacterial resistance to lytic phage infection [19]. We previously described a naturally-occurring phage that leverages this inevitability of phage-resistance evolution as a strength. The dsDNA phage OMKO1 (virus family Myoviridae) attacks *P. aeruginosa*, while affecting the ability for the target bacteria to maintain resistance to various antibiotics [20, 21]. When bacterial strains evolve resistance to phage OMKO1, the mutants can show drug re-sensitivity, suggesting compromised ability for mechanisms such as MexAB and MexXY efflux pumps to remove antibiotics from the cell. Thus, phage OMKO1 can be doubly effective–the virus kills phage-susceptible bacterial cells, and also can drive evolution of phage resistance-associated loss of antibiotic resistance. This 'evolutionary tradeoff' is a mechanistic example of how phages can synergistically interact with chemical antibiotics, to beneficially extend the usefulness of currently-approved drugs. For example, phage OMKO1 and ceftazidime were used successfully in emergency treatment of a 76-year-old patient, to resolve a chronic multi-drug resistant *P. aeruginosa* mediastinal and aortic graft infection [22]. We are also currently testing phage OMKO1 in a clinical trial to resolve or reduce *P. aeruginosa* infections in the lungs of CF, non-CF bronchiectasis and COPD patients when administered via aerosol-delivery (nebulizer) treatment ("CYPHY", NCT04684641).

Since phage OMKO1 has been used successfully in patient treatment, we sought to use it as a model for understanding the stability of phage particles across environmental conditions. Virus particles can become damaged over time, including to the point of total decay (i.e., unable to successfully infect a host cell). The rate of damage can depend on stress imposed by environmental factors, including elevated temperature, and exposure to high concentrations of salt and urea [23–31]. Because damage alters the efficacy of phage infection, and decay alters the concentration of active virus particles, both can create inaccuracies for treatment dosing and administration. In addition, depending on the sensitivity of phage particles to conditions present in the human body, damage and decay could also play a role in the

pharmacokinetics during phage therapy treatment. Thus, it is vital to test how exposure to environmental stressors affects both the titer and activity of phage-therapy candidates. Here, we use phage OMKO1 as a model to identify the effects of three environmental stressors (salt, urea, and heat) on the stability and subsequent infectivity of virus particles. We assess conditions both within and beyond those experienced in the human body, with our findings highlighting the importance of carrying out similar and additional assessments of stability for candidate therapeutic phages.

## Materials and methods

### Strains

*P. aeruginosa* strain PAO1 was provided by B. Kazmierczak (Yale School of Medicine) and cultured on 1.5% agar plates and liquid media made from Lysogeny Broth (LB: 10 g/L tryptone, 10 g/L NaCl, 5 g/L yeast extract). Bacteria were incubated at 37°C, and overnight batch cultures were grown with shaking (100 rpm) in 10mL broth, by inoculating with a single randomly-chosen colony. Bacterial stocks were stored in 40% glycerol at -80°C.

Phage OMKO1 was originally isolated from an aquatic sample (Dodge Pond, East Lyme, CT) enriched on PAO1 [20]. High-titer stocks (lysates) of the phage were obtained by mixing 10μl of the original stock with 10ml of PAO1 culture in exponential phase. After 12 hours to allow phage population growth, the mixture was centrifuged and filtered (0.22μm) to remove bacteria to obtain a cell free lysate. Phage lysates were stored at 4°C. Eight biological replicate OMKO1 stocks (S3, S8, S11, S16, R3, CV1, EV1, EV2) used in some experiments were generated by adding 10μl of the original stock into 5 independent 10ml PAO1 cultures in exponential phase. For simplicity, these were labeled stocks A, B, C, D, E, F, G, and H respectively.

### Measuring bacterial and phage densities

The density of a bacterial suspension was estimated using measurements of optical density at 600nm (OD600), based on a pre-generated standard curve for conversions between OD600 and colony forming units (CFU) per mL in a bacterial culture.

Estimates of phage titers were obtained via classic plaque-assay methods [32]. Lysate samples were serially diluted in LB, and 100 μL of phage dilution were mixed with 100 μL overnight PAO1 culture and added to 4 mL 'soft' (0.75%) agar LB held at 50°C. After gentle vortexing, the mixture was immediately spread onto a 1.5% agar LB plate. When solidified, plates were inverted and incubated overnight at 37°C. Plates with countable plaques were used to calculate the number of plaque-forming-units (PFUs) per mL in the lysate.

### Heat stress survival

To measure effects of heat on phage stability, we exposed samples of phage OMKO1 to temperatures ranging from 55°C to 80°C for 5, 30, 60, or 90 minutes (Fig 1A), or 70°C for 5, 90, 180, 270, or 360 minutes (Fig 1B). For each temperature-duration treatment 100 μL of an OMKO1 phage stock (stock A-E; for Fig 1A, only Stock A) stored in LB media was split evenly into two replicate 250 μL Eppendorf tubes and placed in a pre-heated MyCycler PCR block (BioRad Laboratories, Inc., Hercules, CA) with an established temperature gradient (30–80°C, at 5°C intervals). Tubes were removed from the heat block after specified times, and immediately cooled on ice to halt the stressful condition. The two replicate tubes were mixed together, and then titered in triplicate to measure phage densities. Percent survival was calculated relative to the density measured in the 55°C treatment after 5 minutes (Fig 1A), or relative to the source stock density (Fig 1B). We then fit a generalized linear model and carried out an

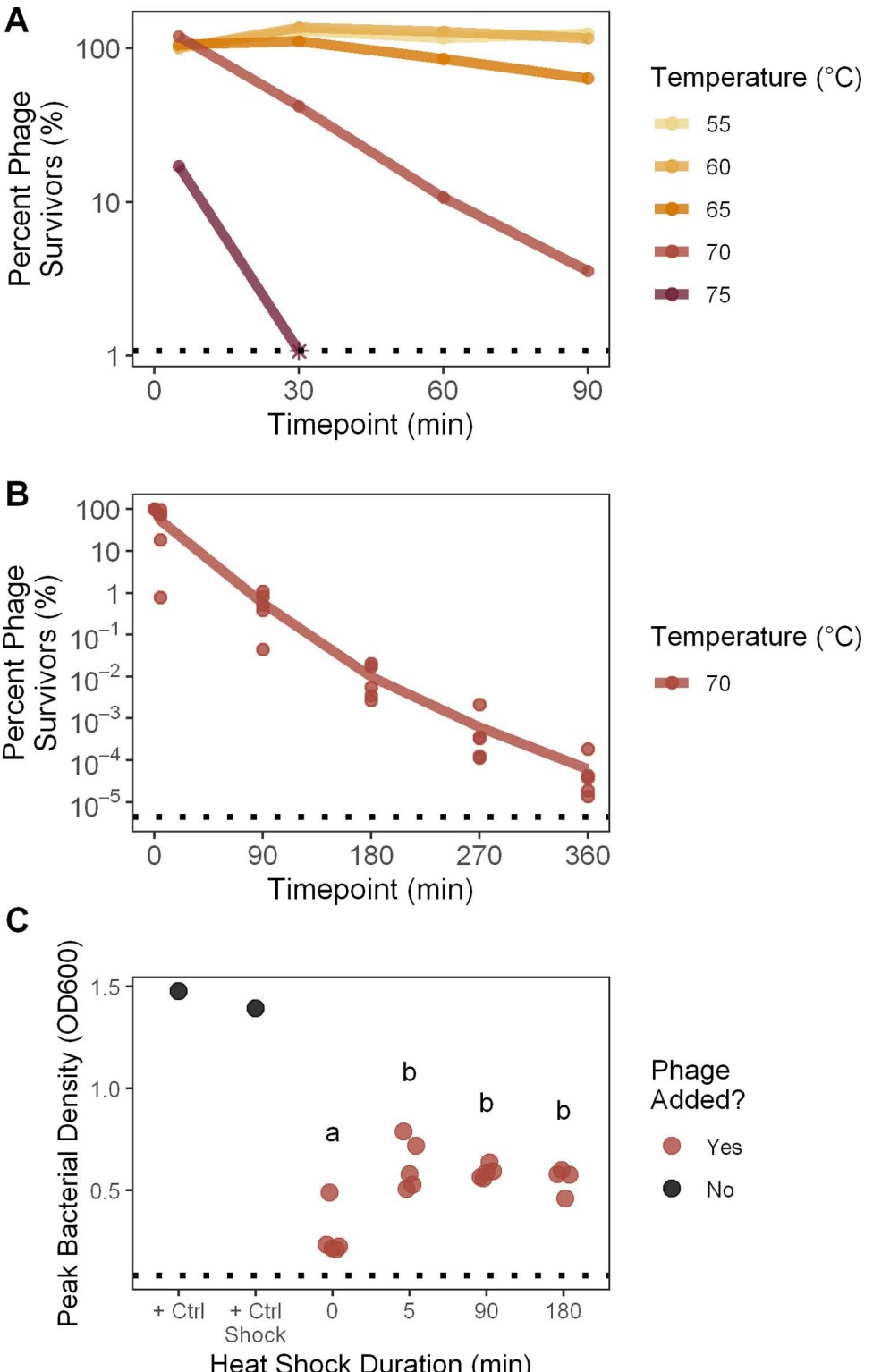

**Fig 1.** Inactivation & fitness suppression of OMKO1 by thermal stress. **A.** To measure phage particle survival of heat stress at different temperatures, OMKO1 was exposed to one of a range of temperatures for 5, 30, 60, or 90 minutes, then titered. Percent survival is plotted relative to the titer of the 55°C treatment after 0 minutes. The dotted line denotes the limit of detection, with the timepoint where survival fell below the limit of detection plotted as an asterisk. 80°C was also tested but caused such rapid particle decay that all measures fell below the limit of detection. **B.**

To measure phage particle survival of heat stress over longer periods of time, five biological replicates of phage OMKO1 were exposed to 70°C for 5, 90, 180, 270, or 360 minutes, then titered. Percent survival is plotted relative to the source stock titer. The dotted line denotes the mean limit of detection across all five batches. **C.** To determine whether phage fitness is affected by a history of heat stress exposure, 70°C heat shocked phages or unshocked control phages (0 min) were inoculated with bacteria and grown overnight while measuring bacterial density. As a metric of phage fitness, the peak bacterial density was computationally determined. Thus, higher peak bacterial densities indicated phages with lower fitness. The dotted line denotes the absence of bacterial growth. Bacteria were also grown in the absence of phage in LB media ("+ Ctrl") or LB media that had been heat shocked for 360 mins at 70°C ("+ Ctrl Shock"). Heat shock treatments that are not significantly different from each other via Tukey Honest Significant Differences are indicated by the same shared letter (a or b).

ANCOVA. For Fig 1A, the model was of $\log_{10}$(percent survival) as a function of temperature (as a factor) and the interaction between temperature and duration; for Fig 1B, of $\log_{10}$(percent survival) as a function of phage stock and the interaction of phage stock and duration. Values below the limit of detection were excluded from statistical analyses.

## Measuring phage fitness by bacterial growth curves

To measure phage fitness post heat stress, phages were subjected to heat stress as described above then titered. These samples were then normalized by concentration to combine a defined number of post-stress PFUs with bacteria in a 200 μL volume of LB in replicate wells (n = 3 or 6) of a 96-well plate (Corning Inc., Corning, NY). For most populations, 200 PFUs were inoculated in each well. However, because of low phage survival rates at longer heat shock times, some wells received fewer PFUs (S3 Fig in S1 File). Across all wells, however, bacteria and phage were inoculated at a constant multiplicity of infection (MOI; ratio of phage particles to bacterial cells) of $10^{-5}$. Plates were incubated >12 hours at 37°C and an automated spectrophotometer (INFINITE F500 microplate reader, TECAN Schweiz AG, Männedorf, Switzerland) was used to monitor changes in the bacterial density (OD600) every 5 minutes, as described above. Growth curves were smoothed and the first local maximum optical density of the bacterial population was used as a reverse proxy for the fitness of phage (S2 Fig in S1 File) [33, 34] (see R script at https://github.com/mikeblazanin/tin-omko1). We then fit a linear model and carried out an ANOVA for peak bacterial density as a function of heat shock treatment (as a factor), phage stock, and treatment-stock interaction. Treatments which were not initially inoculated with 200 PFUs were excluded from statistical analyses. Post-hoc pairwise comparisons between all levels of the heat shock treatment were performed with Tukey's Honest Significant Differences.

## Urea & salt stress survival

To measure the effects of urea (carbamide) and salt (NaCl) on phage stability, 10 μL of phage stock F, G, or H was added to 990 μL of either urea or salt solution at a defined concentration, and vortexed to mix thoroughly. For saline, these included 0M, 0.5M, 3M, 5M, and an LB control (0.17M). For urea, these included 0M, 1M, 2M, 3M, and 4M. One replicate of each of the three phage stocks was exposed to each concentration. At 45 and 90 minutes, subsamples were obtained by removing 50 μL and immediately diluting 200-fold to terminate the stress condition. Then, the stressed samples were titered in triplicate to calculate survival following stress, with percent survival calculated relative to the control titer (for saline, LB; for urea, 0M) at 0 minutes. We then fit a generalized linear model and carried out an ANCOVA of $\log_{10}$(percent survival) as a function of urea or saline concentration (as a factor) and the interaction between concentration and duration. Values below the limit of detection were excluded from statistical analyses.

## Analysis

All analysis was carried out in R (3.6.0) using dplyr (0.8.2), figures were made using ggplot2 (3.2.0) and ggsignif (0.6.0) [35–38]. All data analysis and visualization code is available at https://github.com/mikeblazanin/tin-omko1. Cleaned data and analysis code is also archived at https://doi.org/10.5061/dryad.k3j9kd595.

## Results

To estimate the tolerance of phage OMKO1 to heat stress, we measured phage particle survival over time at different temperatures. Temperature shock significantly increased phage OMKO1 decay rate (Fig 1A, ANCOVA, $F(4, 8) = 125.01$, $p < 0.001$). At the two lowest temperatures (55°C and 60°C), there was no significant evidence of OMKO1 decay over time (Table 1). However, at 65°C there was a significant moderate signal of decay, which became more pronounced at 70°C. At 75°C and 80°C, phage particles decayed so rapidly that most measures in these environments were unobtainable (i.e., fell below the limit of detection). Abnormal plaque morphology was also observed in heat-treated phages, suggesting possible phenotypic alteration of phage OMKO1 following heat treatment, explored further below.

Given the observed substantial decay of phage OMKO1 induced by elevated temperatures, we next sought to observe the dynamics of heat-induced decay over longer periods of time to assess whether decay would continue as exponential or appear biphasic. As expected, we found significant decay over time at 70°C (Fig 1B, ANCOVA effect of duration, $F(1, 20) = 519.1$, $p < 0.001$), and the rate of decay was similar to the prior experiment [Fig 1A slope of $\log_{10}$(percent) at 70°C = -0.0167, Fig 1B slope of $\log_{10}$(percent) of stock A = -0.0161]. We did not observe a notably biphasic decay curve. Surprisingly, biological replicates (phage stocks) had somewhat different decay dynamics, although much of this pattern was driven by a single replicate (ANCOVA effect of stock, $F(4, 20) = 2.54$, $p = 0.07$; ANCOVA interaction between stock and time, $F(4, 20) = 0.95$, $p = 0.45$).

Finally, we sought to test whether exposure to heat might affect the subsequent growth abilities of surviving phage particles. To do so, we took the phage particles that had been heat-shocked at 70°C for different durations of time and normalized their concentrations according to how many plaques they formed. Because of low titers, some populations after 270 and 360 minutes of heat shock could not be normalized, so those timepoints were excluded from analysis (see S3 Fig in S1 File for data). Then, we measured how well these standardized phage suspensions suppressed the growth of host bacteria *P. aeruginosa* cultured for 12 hours (see Methods). As expected, bacterial densities initially increased, before peaking and declining as phages lysed bacterial cells (S2 Fig in S1 File). From these data, we extracted the peak density of bacterial population size as a proxy for gauging phage growth ability (fitness), where lower peak bacterial densities reflected higher fitness of tested phage populations. For comparison,

**Table 1. Multiple regression shows significant phage OMKO1 decay at 65°C and higher temperatures.**

| Temperature (°C) | Estimated Coefficient | t-value | Bonferroni-adjusted p-value |
|---|---|---|---|
| 55 | 0.0008 | 0.98 | 1 |
| 60 | 0.0005 | 0.58 | 1 |
| 65 | -0.0028 | -3.35 | 0.020 |
| 70 | -0.0182 | -22.08 | < 0.001 |
| 75–80 | NA | NA | NA |

Parameter estimates of the rate of decay over time (slope) depending on temperature were evaluated to detect increased decay (decreased slopes) against the null hypothesis of 0 slope using one-tailed t-tests (df = 8) and a Bonferroni correction (4 tests).

we included two controls where bacteria were cultured in the absence of phage, in standard LB medium and in LB medium that was heat-shocked for 360 minutes.

We observed a significant effect of heat-shock exposure on the subsequent fitness of virus particles (Fig 1C), as measured by peak density of the bacterial population (ANOVA, $F_{(3, 15)}$ = 14.2, $p < 0.001$). As expected, the presence of unshocked phage OMKO1 (0 min) reduced peak bacterial density relative to the phage-free bacterial controls, indicating that the phage negatively affected bacterial growth (S1 Table in S1 File). Surprisingly, a history of exposure to heat shock reduced the ability of phage particles to suppress bacterial growth, permitting higher peak bacterial densities (Tukey Honest Significant Differences among all heat stress treatments showed significant differences between unshocked control of 0 minutes and all other treatments at $p < 0.01$, see Fig 1C). However, increasing durations of heat shock exposure did not lead to further decreases in phage fitness (Tukey HSD finds no significant differences among 5 minutes through 180 minutes treatments, $p \geq 0.69$). Unexpectedly, this remained true even after extremely long heat shocks, where the survival rate was very low (e.g. after 180 minutes survival was ~0.01%; Fig 1B). We noted that this increase in peak bacterial density could not be explained by the effect of heat-shocked medium alone, as heat shocked media had the opposite effect of reducing peak bacterial density (+ Ctrl Shock vs + Ctrl, two-sample t-test of unequal variance, $t = -4.18$, $df = 8.86$, $p = 0.002$). These results indicated that heat shock not only inactivated viral particles (Fig 1A and 1B), but also reduced the subsequent growth of surviving phage particles (Fig 1C).

Next, to estimate the tolerance of phage OMKO1 to saline stress, we measured phage particle survival over time in different saline concentrations, relative to starting densities. Although there was a significant effect of saline concentration on densities over time (slope of the lines in Fig 2; ANCOVA: $F_{(5, 23)} = 5.57$, $p = 0.002$), higher saline concentrations did not accelerate

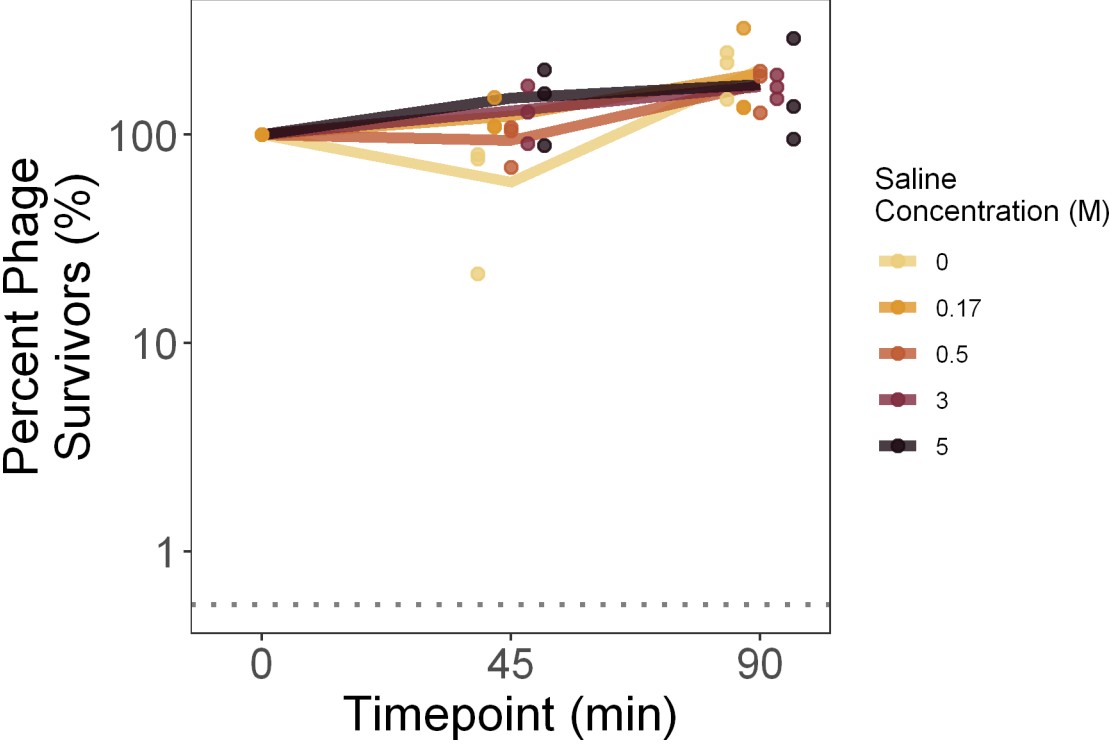

**Fig 2.** Phage OMKO1 decay is not accelerated by saline concentration.

phage decay (S2 Table in S1 File). This was likely because of the unexpected increase of phage densities in some treatments, which we ascribed to sampling variation. These findings were consistent with other data collected on a single phage stock with greater sampling density, where saline had no effect on the decay rate of phage OMKO1 (S4 Fig in S1 File).

To measure phage particle survival of saline stress, phage OMKO1 was exposed to a range of saline concentrations then titered after 45 and 90 minutes. Percent survival is plotted relative to the titer of the LB control (0.17 M) at 0 minutes. Bold lines denote the average of the three biological replicates, with individual replicates plotted as points (horizontally offset for visualization). The mean limit of detection between the replicates is plotted as a dotted line.

To estimate the tolerance of phage OMKO1 to urea stress, we measured phage particle survival over time in different urea concentrations, relative to starting densities. Higher urea concentrations elevated the rate of decay of phage particles (Fig 3), although this trend was not significant after measures below the limit of detection were removed (slope of the lines in Fig 3; ANCOVA: $F_{(5, 15)} = 1.19$, $p = 0.36$, S3 Table in S1 File). In particular, within the 90-minute assay, we observed that phage survival was unaffected by urea concentrations up to 2M, while phages decayed more rapidly in 3M and 4M. These findings were consistent with data collected on a single phage stock with greater sampling density, although there urea significantly increased decay at 1M and above (S5 Fig in S1 File, S4 Table in S1 File).

To measure phage particle survival of urea stress, phage OMKO1 was exposed to a range of urea concentrations then titered after 45 and 90 minutes. Percent survival is plotted relative to the titer of the control (0 M) at 0 minutes. Bold lines denote the average of the three biological replicates, with individual replicates plotted as points (horizontally offset for visualization). Points which fall below the limit of detection are plotted at the limit of detection for that batch as asterisks.

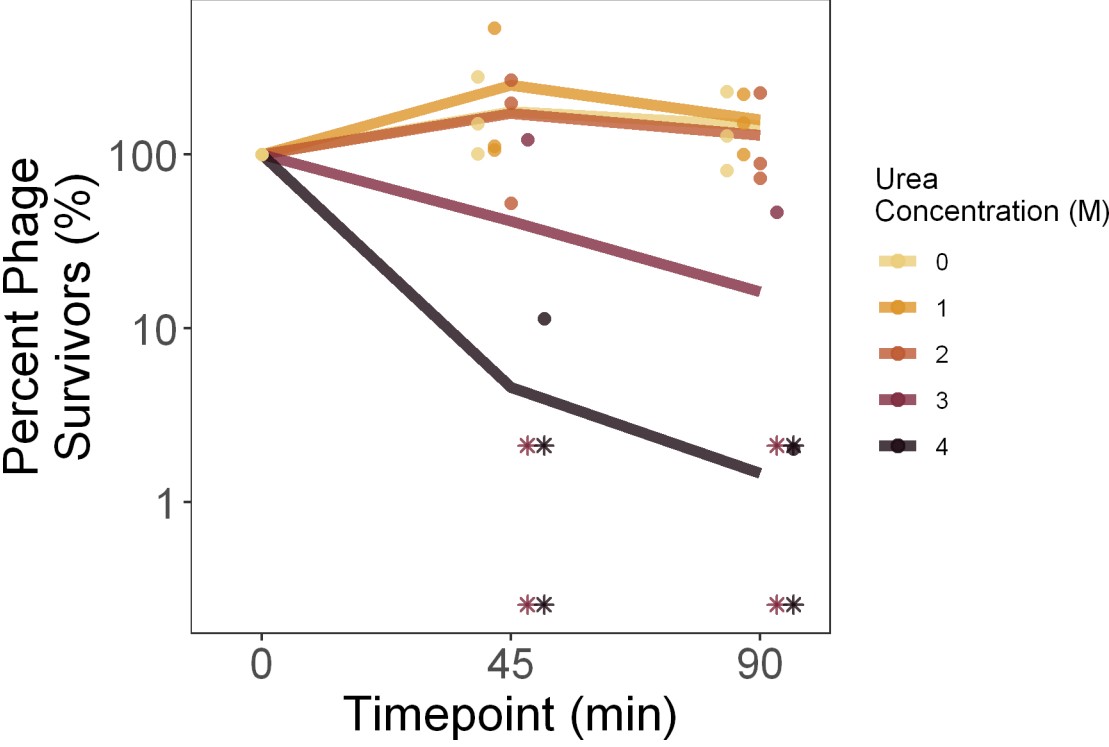

**Fig 3.** Degradation of phage OMKO1 under different urea concentrations.

## Discussion

With the rise of antibiotic resistant bacterial pathogens, there is increasing interest in phage therapy to complement or replace traditional chemical antibiotics. However, phage virions are susceptible to environmentally-induced damage and decay from environmental stressors. Here, we used the decay and damage of therapeutic phage OMKO1 by salt, urea, and heat as a model of the physiological limits of phage particles. We observed no measurable adverse effect of salinity on phage OMKO1 survival (Fig 2), but strong effects of both increased urea concentrations (Fig 3) and elevated temperatures (Fig 1A and 1B) on virus survival. In addition, we found that phage particles which survived elevated temperatures had reduced fitness, as measured by their ability to suppress the growth of susceptible bacterial cells (Fig 1C).

Exposure to high temperatures caused both decay and damage of phage. Virus particles decayed approximately exponentially (Fig 1B, linear on log-linear axes), with higher temperatures increasing the rate of degradation (Fig 1A), although we have limited resolution to detect a bi-phasic decay, as observed elsewhere [26]. Intriguingly, OMKO1 particles which survive exposure to heat stress experience a drastic reduction in fitness. This effect is apparent even after only five minutes of exposure, and does not become more pronounced following longer durations of heat stress (Fig 1C). Along with the observed abnormal plaque morphology after heat shock, these results indicate a strong plastic response of phage fitness to high temperature exposure. We propose three possible explanations for this observation. The first is that the population of OMKO1 particles is heterogenous in resistance to heat shock, and that stability trades off with fitness. While this sort of stability-function tradeoff is thought to be common for proteins [39, 40], this explanation is incompatible with the observation of simultaneous high survival and large decreases in fitness for two of the replicate populations (e.g. populations A and E after 5 minutes, S1 and S3 Figs in S1 File). The second explanation is that OMKO1 particles which no longer form plaques due to heat stress may inhibit the infection of cells, possibly by competitively binding to phage receptors and blocking infection by "active" phage particles. Although this can explain improved growth of bacteria in the presence of heat-shocked phages, it cannot explain the observed changes when phages have both high survival percentages and low fitness (e.g. populations A and E after 5 minutes, S1 and S3 Figs in S1 File). As a final explanation, we propose that high temperatures alter viral particle conformations to a lower-fitness state, an environmentally-caused damage. Thus, any exposure to heat shock alters the state and phenotype of all surviving viral particles, consistent with the observed changes in fitness following heat shock (Fig 1C). While such bistability is observed here following environmental conditions, phage particle bistability elsewhere has been previously reported to promote the evolution of novel host use [41].

We also observed that phage OMKO1 was deactivated by elevated urea concentrations, consistent with previously-published findings on other phages. For example, some of the earliest work published on this topic tested the survival of particles of coliphages (viruses that specifically infect *Escherichia coli*) over time in a 4.61M urea solution, observing decay anywhere from 0 to 7 orders of magnitude within 30 minutes [28]. Similarly, phage T4 experienced 90% decay after one minute of exposure to 2.5M urea [29]. By comparison, phage OMKO1 decayed more slowly (~95% after 45 minutes at 4M, Fig 3), suggesting that this virus had relatively greater tolerance for elevated urea concentrations compared with the limited evidence from other published studies on phage survival.

In contrast to our findings with urea, our failure to find an effect of saline stress diverged from previously-published results on other phages. Phages T2, T4, T6 and $ps_1$ have been reported to be deactivated when they are rapidly osmotically shocked from 4M to 0M NaCl, and several phages decay in 0M NaCl, all conditions where we observed

no effect [27, 30, 31]. Experimental differences may have contributed to these divergent results, including differences in the rate of dilution and thus severity of osmotic shock, or in the relatively shorter time-lengths of our experiments. Additionally, our findings may reflect some degree of underlying differences in susceptibility to osmotic shock between different phages, where OMKO1 is more tolerant of osmotic shock and saline stress than other reported phages.

Our results are also relevant to the understanding of how environmental stressors might affect use of phage OMKO1 in human therapy. Our experiments were generally designed to exceed the range of environmental conditions that phage particles could experience during storage or within therapy applications in the human body. For example, within the human body the concentrations of salt range from roughly 30mM in urine to 135–145 mM in blood [42], all conditions where we saw no increase in phage OMKO1 decay rate (Fig 2); similarly urea concentrations range from 2.6–6.5mM in human blood up to 325mM in human urine [43], again conditions where we observed no increase in phage decay rate (Fig 3). Human body temperatures can reach a maximum of ~40°C, while phages could experience up to 50°C during storage and transportation. We found that phage OMKO1 shows no significant decay over 90 minutes at 55°C (Fig 1A), suggesting that heat stress is only relevant over these timescales at temperatures well beyond those experienced by a therapeutic phage. This, however, does not rule out the possibility of accelerated decay at lower temperatures from factors others than heat stress. These indications of the general stability of phage OMKO1 despite possible environmental stressors are promising for its further potential uses in clinical applications, although much further characterization is needed to completely determine the stability of OMKO1 against relevant environmental stressors for therapeutic application.

Future studies on the stability of phage OMKO1 and other therapeutic phages should expand the types of examined stressors, consider interactions between multiple stressors, and begin to more deeply elucidate the nature of the observed phage particle bistability. For instance, future work could expand on the scope of this study by considering the effects of pH and electromagnetic radiation, as well as interactions among any environmental stressors or between these stressors and storage conditions [e.g. freezing, lyophilizing (freeze drying)]. Additionally, biophysical and imaging approaches could be utilized to understand the mechanistic details behind phage particle stability and conformation, like those underlying the observed changes in phage fitness after any duration of heat stress exposure. While these experiments would deepen our understanding of phage biology and stability, our current work has revealed the limits of the stability of phage OMKO1 and highlighted intriguing non-linear responses of phages to environmental stress, confirming the need for careful characterization and storage of therapeutic phages for widespread clinical use.

## Supporting information

**S1 File. Supplemental material including statistical tables and additional data figures.** This supplemental file includes further details on statistical analyses of heat, urea, and saline stress assays reported in the main text. Additionally, it includes visualization and statistical analyses of additional data from heat, urea, and saline stress assays not included in the main text. (DOCX)

## Acknowledgments

We thank Alita Burmeister, Caroline Turner, Michael Wiser, and four anonymous reviewers for helpful feedback on the manuscript.

## Author contributions

**Conceptualization:** Wai Tin Lam, Benjamin K. Chan, Paul E. Turner.

**Formal analysis:** Michael Blazanin.

**Investigation:** Michael Blazanin, Wai Tin Lam, Eli Vasen, Benjamin K. Chan.

**Methodology:** Wai Tin Lam.

**Resources:** Paul E. Turner.

**Supervision:** Benjamin K. Chan, Paul E. Turner.

**Visualization:** Michael Blazanin.

**Writing – original draft:** Michael Blazanin, Wai Tin Lam.

**Writing – review & editing:** Michael Blazanin, Wai Tin Lam, Paul E. Turner.

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
