## [Decision Letter · Decision Letter 0]

5 Oct 2021

PONE-D-21-21818

Decay and damage of therapeutic phage OMKO1 by environmental stressors

PLOS ONE

Dear Dr. Blazanin,

After careful consideration and a too long and unsucessful search for appropriate reviewers, we feel that it has merit but does not fully meet PLOS ONE’s publication criteria as it currently stands. Therefore, we invite you to submit a revised version of the manuscript that addresses the points raised during the review process.

ACADEMIC EDITOR:

Please answer to the different technical issues raised by reviewer.

We look forward to receiving your revised manuscript.

Kind regards,

Cecilio López-Galíndez

Academic Editor

PLOS ONE

2.We note that the grant information you provided in the ‘Funding Information’ and ‘Financial Disclosure’ sections do not match.

“support for Wai Tin Lam from the Chinese University of Hong Kong.”

“We disclose financial support for Wai Tin Lam from the Chinese University of Hong Kong.”

“Paul E. Turner discloses a financial interest in Felix Biotechnology, Inc., a phage

therapeutic company with first rights to use patents resulting from this work. Paul

Turner sits on the Board of Directors of Nextbiotics.”

Additional Editor Comments (if provided):

Reviewers' comments:

Reviewer's Responses to Questions

**Comments to the Author**

1. Is the manuscript technically sound, and do the data support the conclusions?

Reviewer #1: Partly

2. Has the statistical analysis been performed appropriately and rigorously? 

Reviewer #1: I Don't Know

3. Have the authors made all data underlying the findings in their manuscript fully available?

Reviewer #1: Yes

4. Is the manuscript presented in an intelligible fashion and written in standard English?

Reviewer #1: No

5. Review Comments to the Author

Reviewer #1: The manuscript “Decay and damage of therapeutic phage OMKO1 by environmental stressors” by Blazanin et al., deals with the analysis of the tolerance ranges of the virulent phage OMKO1 referred to temperature, salinity and urea parameters. Although data included in this study are undoubtedly important, they do not constitute a complete piece of work to guarantee its potential use as therapeutic phage to combat certain infectious diseases due to Gram-negative pathogens.

The above statement is supported by these specific points:

• Purification and characterization of new phages has been the subject of many articles for several decades. Over time, the focus of these analyses has shifted towards molecular techniques and their possible impact on the virulence of the targeted bacteria. Regarding the phages eventually used in therapy, the characterization of tolerance parameters against stress factors is necessary, but not sufficient, to conclude that a certain phage can be tested in in vivo assays.

• The current manuscript contains valuable information on the relevant tolerance of OMKO1 to a wide range of salt concentrations whereas is affected by high concentrations of urea and elevated temperatures. These latter conditions, however, are not totally significative since some of the urea concentrations and temperatures over 55ºC are simply nonsense, in terms of environmental conditions.

• One surprising issue in the manuscript is the comment about the preparation of “high-titer stocks (lysates)” of OMKO1. Does it mean that the phage is not further purified, normally through a CsCl gradient step? This protocol, or something similar, is practically mandatory to obtain highly purified phage with the highest possible titer. In this sense, what does it mean “high-titer” expressed as PFU/mL? In addition, how long are the phage lysates stored at 4ºC? Is this storage in LB medium, as stated in line 123, without any preservative compound? Have the authors checked the evolution of phage titer over time at 4ºC, -20ºC, and -80ºC with glycerol? Does this phage retain its titer with lyophilization and further resuspension? These details are important for the study of any new phage and are lacking in the manuscript.

• Once set up the optimal parameters to retain the highest titer, after checked the storage conditions and the tolerance to different stressors, some in vitro experiments are necessary to carry out, particularly when the phage is thought to use it as “therapeutic” to kill susceptible bacterial pathogens. In this case, the targeted bacterium is apparently restricted to Pseudomonas aeruginosa, an important Gram-negative pathogen that causes varied infections, including those affecting lung in cystic fibrosis patients. These latter infections are well known that are produced by P. aeruginosa grown as biofilms. For these reasons, some crucial experiments would be to check whether OMKO1 is capable of reducing the P. aeruginosa viable cells after adding the phage, both in planktonic and biofilm cultures.

• Given the rationale of the successive and necessary in vitro assays before reaching the validation in in vivo experiments, or clinical trials on patients, it is rather surprising the comments included in the Introduction section of the manuscript (lines 73-77):

We are also currently testing phage OMKO1 in a clinical trial to resolve or reduce P. aeruginosa infections in the lungs of CF, non-CF bronchiectasis and COPD patients when administered via aerosol-delivery (nebulizer) treatment.

While phage OMKO1 already has been used successfully for patient treatment, its stability across environmental conditions remains uncharacterized.

If I understand correctly, phage OMKO1 has been used in this kind of assays or clinical trials without a complete characterization of stability or other in vitro experiments.

6. PLOS authors have the option to publish the peer review history of their article (what does this mean?). If published, this will include your full peer review and any attached files.

Reviewer #1: No

---

## [Author Response · Author response to Decision Letter 0]

26 Oct 2021

See attached Response to Reviewers file for specific responses to all reviewer and editor comments

---

## [Decision Letter · Decision Letter 1]

31 Jan 2022

Decay and damage of therapeutic phage OMKO1 by environmental stressors

PONE-D-21-21818R1

Dear Dr.Blazanin,

We’re pleased to inform you that your manuscript, after a too long review process that I apologize, has been judged scientifically suitable for publication and will be formally accepted for publication once it meets all outstanding technical requirements.

Kind regards,

Cecilio López-Galíndez

Academic Editor

PLOS ONE

Additional Editor Comments (optional):

Take note that one of the reviewers has raised comments about the inclusion of other experiments in the reviewed version that would have improved the quality of the manuscript.

Reviewers' comments:

Reviewer's Responses to Questions

**Comments to the Author**

1. If the authors have adequately addressed your comments raised in a previous round of review and you feel that this manuscript is now acceptable for publication, you may indicate that here to bypass the “Comments to the Author” section, enter your conflict of interest statement in the “Confidential to Editor” section, and submit your "Accept" recommendation.

Reviewer #1: All comments have been addressed

Reviewer #2: All comments have been addressed

2. Is the manuscript technically sound, and do the data support the conclusions?

Reviewer #1: No

Reviewer #2: Yes

3. Has the statistical analysis been performed appropriately and rigorously? 

Reviewer #1: Yes

Reviewer #2: N/A

4. Have the authors made all data underlying the findings in their manuscript fully available?

Reviewer #1: No

Reviewer #2: Yes

5. Is the manuscript presented in an intelligible fashion and written in standard English?

Reviewer #1: Yes

Reviewer #2: Yes

6. Review Comments to the Author

Reviewer #1: The authors have tried to explain the changes made in the revised version on the basis that the main objective of the study is “to test the tolerance limits of OMKO1 to temperature, salinity, and urea (what the authors call "the physiological limits"), serving as a general model for understanding the physiological limits of phage particles”. Thus, comments and suggestions on some experiments required to complete and reinforce the interest of the OMKO1 phage study have been avoided and answered by the authors with the sentence "…they are beyond the scope of the current study and will be published elsewhere".

In the opinion of this reviewer, the current version of the manuscript is simply correct from the point of view of the characterization of some parameters of the OMKO1 phage, but these data are only partial because the suggested experiments that are fully related to the more complete characterization of the OMKO1 phage should be included.

The content of the manuscript in its present version is of little interest to the broad audience on phage readers, and in particular to the implications of OMKO1 for phage therapy.

Reviewer #2: (No Response)

7. PLOS authors have the option to publish the peer review history of their article (what does this mean?). If published, this will include your full peer review and any attached files.

Reviewer #1: No

Reviewer #2: No

---

## [Editor Report · Acceptance letter]

14 Feb 2022

PONE-D-21-21818R1 

Decay and damage of therapeutic phage OMKO1 by environmental stressors 

Dear Dr. Blazanin:

I'm pleased to inform you that your manuscript has been deemed suitable for publication in PLOS ONE. Congratulations! Your manuscript is now with our production department. 

Kind regards, 

on behalf of

Dr. Cecilio López-Galíndez 

Academic Editor

PLOS ONE